

# Optimized photodegradation of Bisphenol A in water using ZnO, TiO₂ and SnO₂ photocatalysts under UV radiation as a decontamination procedure

Rudy Abo[1], Nicolai-Alexeji Kummer, Broder J. Merkel

[1]Department of Geology, TU Bergakademie Freiberg, Freiberg, 09599, Germany

*Correspondence to*: Rudy Abo (rudy.abo@hotmail.de)

**Abstract.** Experiments on photodegradation of Bisphenol A (BPA) was carried out in water samples by means photocatalytic and photo-oxidation methods in the presence of ZnO, TiO₂ and SnO₂ catalysts. The objective of this study was to develop an improved technique that can be used as remediation procedure for BPA contaminated surface and groundwater based on solar

radiation. The photodegradation of BPA in water was performed under low-intensity UV mimics natural solar radiation. The results reveals significantly higher degradation rates observed in the presence of ZnO than with TiO₂ and SnO₂ catalysts during 20 h of irradiation. The intervention of the advanced photocatalytic oxidation (PCO) was reduced the time of degradation to less than 1 h to reach a degradation rate of 90% for BPA in water. The results also suggests the use of ZnO as competitor catalyst to the traditional TiO₂, providing most effective treatment of contaminated water with phenolic products.

**Keywords:** Bisphenol A (BPA), Photodegradation, Catalyst, Photocatalytic oxidation (PCO), Endocrine disruptor, UV irradiation

## 1 Introduction

Bisphenol A (BPA; 4,4´-isopropylidenediphenol; Fig. 1) is a common industrial product, manufactured in large quantities worldwide. Almost 99% of the BPA are used as a common intermediate compound in the production of polycarbonate, epoxy

resins, unsaturated polyesters and styrene resins (Staples et al., 2000). BPA is used as a coating material for cans, and as additive in powder paints, thermal paper, dental fillings, antioxidants and polycarbonate plastic as well. Therefore, it is usually found in many plastic products including water pipes, drinking water containers and tableware (Vandenberg et al., 2007).

Recent studies have shown that BPA has potentially toxic effects, inducing estrogenic endocrine disruption and increased risk of tumorigenesis (Keri et al., 2007; Vandenberg et al., 2007). They also indicated that BPA could trigger the disruption of

corpuscular function even at very low concentration (0.23 ng/l) leading to disorders of estrogenic hormone secretion (Chiang et al., 2004). BPA is released into water resources not only by discharge from manufacturing, but also from contaminated drainage networks, sewage and landfills (Welshons et al., 2006). In particular in developing countries BPA is produced on the one side, but on the other side, without proper measures in operation regulating and monitoring the discharge of BPA containing fluid and solid waste into the environment and in particular aquatic systems (Arukwe et al., 2012; Baluka and

Rumbeiha, 2016). Disintegrated BPA derived from various industrial products under different weather conditions could reach



surface and groundwater by leaching processes. Furthermore, BPA can easily enter the human body producing a broad spectrum of adverse health effects (Zhang et al., 2006). In fact, purification of contaminated surface and groundwater has become a priority for some countries, a strategy that is in line with the increasing global demand for fresh water. Photodegradation can provide an affordable enhanced natural attenuation solution and add to the arsenal of classic treatment
techniques.

Photodegradation primarily occurs by either photolysis or photo-oxidation. Photolysis is a process in which chemical compounds dissolved in water (e.g. phenols and BPA) absorb sunlight or other illumination sources which directly results in their photochemical alternation (Howard, 1991a), while photo-oxidation involves the degradation of compounds through interaction with hydroxyl radicals or other similar oxidants. Both reactions can occur naturally in water and in the atmosphere
(Staples et al., 1998). Moreover, disintegration of contaminants by photodegradation also can be explained by the photo-Fries rearrangement of BPA polycarbonate (Rivaton, 1995). The path of degradation is shown in the Fig. 2. Photo-Fries is complex reaction occurs as combined or independent process, and plays most important role during the outdoor weathering processes of polycarbonate (Pickett, 2011). The exposure of BPA with short wavelengths leads to substantial rearrangement of the aromatic carbonate into phenylsalicylate and dihydroxybenzophenone derivatives (Diepens and Gijsman, 2007).

Lemaire, et al., proved that photo-Fries rearrangement reactions are more likely to occur when short UV wavelength are used (<300 nm). In contrast, advanced degradation (e.g. photo-oxidation) reactions have a high efficiency when wavelengths >340 nm are used (Rivaton et al., 1983). However, BPA exhibits absorption of UV wavelength exceeding 290 nm in neutral and acidic methanol solutions (Diepens and Gijsman, 2007; Howard, 1991b). Metal oxides have been widely used as catalysts for photodegradation in recent years and can be used for the successful treatment of water and total oxidation of organic pollutants
considering the type and magnitude of contamination (Ferrandon, 2001; Ibhadon and Fitzpatrick, 2013; Pelizzetti et al., 1993). Generally, the effectiveness of a photocatalyst is characterized by its capacity to simultaneously adsorb reactants and photon energy. This is reflected by the photosensitivity of the catalyst and its large band gap (Du et al., 2009; Sakthivel et al., 2003). Photon energy greater than the semiconductor photocatalyst band gap (wavelength 254 nm) creates electron hole pairs (Horikoshi et al., 1998; Kaneco et al., 2004). The hole pair ($e^-$-$h^+$) is generated at the surface of photocatalyst as shown in the
equations below:

$$photocatalyst + hv \rightarrow e_{cb}^- + h_{vb}^+ \qquad (1)$$

$$h_{vb}^+ + H_2O \rightarrow \cdot OH + H^+ \qquad (2)$$

$$e_{cb}^- + O_2 \rightarrow \cdot O_2^{\cdot -} + H^+ \rightarrow \cdot OOH \qquad (3)$$

Where; $hv$ is the photon energy, $e_{cb}^-$ is the valence electron which is released to conduct the catalyst band and leaves the hole
$h_{vb}^+$ in the valence band.

The hydroxyl radicals (•OH) are powerful oxidants that transform BPA into different forms of phenolic group compounds (Abdollahi et al., 2011). Furthermore, the advanced photocatalytic oxidation using sodium hypochlorite (NaOCl) as an oxidizing agent can accelerate the degradation efficiency by releasing oxygen ($O_2$) into the water. The decomposition of NaOCl into hydrochloric acid and oxygen occurs when exposed to UV irradiation as follows (Fukuzaki, 2006; Scanlon et al., 2002):



$$NaOCl + H_2O \rightarrow HOCl + NaOH \qquad (4)$$

$$2HOCl \rightarrow 2HCl + O_2 \qquad (5)$$

Oxygen as well is a very strong oxidant. Therefore, it disintegrates BPA into $CO_2$ and water $H_2O$ (Kaneco et al., 2004):

$$C_{15}H_{16}O_2 + 18O_2 \rightarrow 15CO_2 + 8H_2O \qquad (6)$$

One of the most popular catalysts, $TiO_2$, is widely used for decontamination of various organic contaminants in water due to its attractive physical properties. Few studies have dealt with other catalysts such as ZnO and $SnO_2$, particularly the ZnO which shows high stability and a large band gap in comparison to other existing catalysts. The objective of this study was to optimize the treatment procedure accelerating the photodegradation process of BPA and other phenolic compounds in natural water by using various photodegradation approaches; photodegradation, photo-oxidation/ photocatalytic degradation and advanced

photocatalytic oxidation degradation.

## 2 Materials and methods

The treatment approach was optimized through different stages considering various factors affecting photodegradation. High purity bisphenol A (99.99%), acetonitrile (HPLC grade >99.99%) and sodium hypochlorite NaOCl (14~16%) were purchased from Merck (Darmstadt, Germany). The concentration of NaOCl is determined by titration with sodium thiosulfate. Ultra-pure

water was provided using TKA MicroLab UV system (Niederelbert, Germany). High purity ZnO and $SnO_2$ powder grade catalysts (> 99.999%; powder < 5 µm) were purchased from Alfa Aesar (Karlsruhe, Germany) and $TiO_2$ rutile grade (> 99.999%; powder < 5 µm) from Sigma Aldrich (Hamburg, Germany). Water samples filtration was performed using 0.45 µm polytetrafluoroethylene filter (Restek). BPA was analyzed by HPLC equipped with an L-4250 UV-VIS detector (Merck-Hitachi D6200A, Tokyo, Japan) at a wavelength of 254 nm using a Nucleosil C18-EC 125/4 column (Machery-Nagel GmbH;

Düren, Germany). Analyses were performed isocratically at a flow rate of 1 ml/min and a 20 µl sample loop. As mobile phase a mixture of acetonitrile and water (70:30) was used. The quantification was carried out by external calibration (linear regression) with seven standard solutions in the range 5 to 60 ppm prepared using a 100 ppm BPA stock solution in acetonitrile solvent. Each of the BPA standards were prepared in 25 ml distilled water. Since the UV absorbance range is one of the most important factors affecting photodegradation, the maximum absorption value of BPA in aqueous samples was investigated

using an Ocean Optics USB2000+ spectrometer (Dunedin, USA) equipped with a Deuterium Tungsten Halogen light source (DH-2000). 1.5 ml of 25 ppm BPA solution prepared in water was filled in a 2 ml optical cell and analyzed in the wavelength range of 220-400 nm.

### 2.1 Photodegradation under UV

Photodegradation of 25 and 50 ppm BPA in 100ml aqueous samples was achieved using low pressure UV illumination

provided by Köhler NU-6KL (Neulußheim, Germany) equipped with two UV lamps. These lamps deliver two different wavelengths, 254 and 365 nm, and an intensity range from 600-800 µW/cm³. The BPA samples in 250 ml rectangular glass



beakers were exposed to direct UV irradiation (254 nm wavelength) with a distance of about 5 cm between the aqueous phase and the UV source. During the experiment, 5 ml of water samples were taken every 60 min. 100 µl of the filtered samples containing BPA were injected into the HPLC device by using a 250 µl glass syringe in order to observe the changes in concentration of BPA over time and the effect of the initial concentration on photodegradation. The above mentioned HPLC

procedure was used in the experiment.

## 2.2 Photocatalytic photodegradation

Photocatalytic degradation was used as more advanced decontamination technique to remove BPA from water sample. Suspensions of the catalysts ZnO, $TiO_2$ and $SnO_2$ (each with 0.1 %( w/w)) and 25 ppm BPA in 100 ml ultra-pure water were prepared and mixed for 15 min using a magnetic stirrer at 400 rpm rotation speed in order to allow maximal sorption of BPA

on the catalysts' surface. Then the suspensions of the catalysts were directly irradiated for 5 h using both wavelengths (254 and 365nm). Changes in BPA concentration were measured every 60 min using HPLC. Since the catalysts' amount is an important factor for photodegradation, the influence of catalysts' concentration on the photodegradation efficiency was investigated using two different concentrations of 0.1 and 0.2% (w/w).

## 2.3 Advanced photocatalytic oxidization

Advanced photocatalytic oxidization (PCO) experiments were carried out using a relatively low concentration of sodium hypochlorite (NaOCl) as an oxidizing agent in the presence of photocatalysts. Photodegradation was studied using suspensions of 25 ppm BPA in 100 ml ultra- pure water, 0.1% (w/w) of ZnO, $TiO_2$, $SnO_2$ catalysts and 0.3 mM NaOCl. The samples were mixed for 15 min and placed under direct UV irradiation using both wavelengths 254 and 365 nm. Changes in BPA

concentration were measured every 15 min during the total irradiation time of 1 h. Furthermore, the influence of different amounts of NaOCl on the photodegradation of BPA was investigated using ZnO as photocatalyst and 0.1, 0.3 and 0.5 mM of NaOCl, respectively. 100 µl of the filtered aqueous samples were analyzed by HPLC at each time interval.

## 2.4 Degradation products

Photodegradation products of the advanced photocatalytic oxidation experiments were investigated using Thermo Scientific Trace GC-MS (Bonn, Germany) equipped with a TG-1MT capillary column (30 m x 0.25 mm x 0.25 µm). For the experiments, suspension of 25 ppm BPA, 0.1% (w/w) ZnO catalyst and 0.1 mM NaOCl in 500 ml ultra-pure water were prepared in rectangular glass beakers. 50 ml of the suspension were taken during the photodegradation experiment every 15 min. They were pre-concentrated using solid phase extraction, SPE (Fig. 3). This was performed using 3 ml Chromabond HR-P glass

cartridges purchased from Macherey-Nagel (Düren, Germany) filled with 200 mg of a highly porous polystyrene-divinylebenzene copolymer with high binding capacity up to 30 % of the adsorbent weight. The extraction of aqueous samples

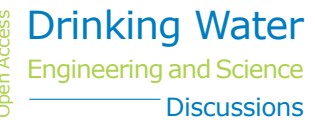

was carried out using the following procedure: SPE cartridges were conditioned using 3 ml of methanol followed by 3 ml of water. 50 ml water samples were applied into the cartridges at a flow rate of 15 ml/min. 3 ml of water adjusted to pH 2.0 using 0.1 M HCl was used in the washing stage. Then organic compounds were extracted using 3 ml of high-purity acetonitrile as final step. 1 µl of the finally extracted samples were then injected into the GC-MS device in splitless mode. The injection temperature was 260°C. The oven temperature was adjusted as follows: initial oven temperature was 120°C for 4 min, then it was increased to 250°C with 30 °C/min followed by a hold time of 1 min, then the temperature was increased to 270°C at 20 °C/min and held for 3 min. The transfer line temperature was adjusted to 300°C. Helium was used as carrier gas with a flow rate of 1.2 ml/min. The temperature of the ionization source was adjusted to 300°C. Qualitative analysis was achieved using electron ionization at 70eV and scan mode range from 45-400 m/z.

## 3 Results and discussion

### 3.1 UV-VIS absorbance spectrum

The UV spectrum of BPA in aqueous solution was measured as illustrated in Fig. 4. The results showed significant UV absorption at wavelengths greater than 250 nm. The spectrum showed a high optical density (OD) in the range of 250–290 nm with ʎ$_{max}$ at 265 nm, confirming previously published findings (Diepens and Gijsman, 2008). Therefore, two ranges of UV irradiation were used to insure maximum absorption to accelerate the disintegration reactions.

### 3.2 BPA photodegradation under UV

The results show a significant decrease in BPA concentration in the aqueous samples (Fig. 5). This accompanied with color changes of the solution to yellow and light greenish-yellow as a result of the disintegration of BPA into other derivative products which could be an indicator of the degradative progress. The change in pH was investigated during the experiment and it showed a decrease during the experimental time ranging from 0.3 to 0.5 units below the initial value (pH=5.8). This can be explained by the increase in carbonic acid ($H_2CO_3$) concentration resulting from dissolved $CO_2$ in water (Donald, 1997). The photodegradation was relatively slow within the first 4 h of the experiment, but thereafter occurred rapidly. The increased breakdown after 240 min of irradiation could be explained by the photo-Fries rearrangement of BPA which increases the photo-oxidation rate (radicals forming), and thus accelerates the degradation reaction (Diepens and Gijsman, 2007). This process breaks the bonds between the atoms of BPA at specific moments and produces simpler organic compounds that can be easily disintegrated.

### 3.3 Influence of BPA initial concentration

The results also showed that the lower the initial concentration of BPA, the higher the degradation rate (Fig. 6). The concentration of bisphenol A in the two samples decreased gradually after 10 h of degradation. Nevertheless, the degradation



rate of BPA in the 25 ppm solution was 20% higher than that of the 50 ppm solution resulting from higher disintegration efficiency of the solution of lower concentration.

### 3.4 Photocatalytic degradation of BPA

Higher degradation rate was achieved when ZnO suspension was used rather than one of the other catalysts (Fig. 7), so that 98% of the BPA were degraded after 5 h. The results showed also that ZnO seems to be a suitable alternative to TiO$_2$ that is commonly used. ZnO shows high efficiency during degradation, and has a wider band gap ($\Delta_{ZnO}$ = 3.4) and higher photosensitivity compared to other semiconductors ($\Delta_{TiO2}$ = 3.2, $\Delta_{SnO2}$ = 3.3) (Thomazi et al., 2009; Yang et al., 2006).These properties make the ZnO catalyst more susceptible to absorbing UV light below 380 nm. The GC-MS results for already degraded samples showed concentrations of both butylated hydroxytoluene and chlorohydroquinone as the major degradation byproducts at 88% confidence. The butylated hydroxytoluene (BHT) is characterized chemically as a derivative of phenol and used primarily as additive for food and cosmetics due to its antioxidant properties. It is non-toxic at intake rates above 50 mg/kg/day (Branen, 1975). Chlorohydroquinone exposure has been listed in the classification of the National Fire Protection Association (NFPA) standard system for the identification of the hazards of materials for emergency response at the level 1 on a scale from 0 (no hazard) to 4 (severe risk) (NFPA, 2011).

### 3.4 Effect of the amount of photocatalyst

The efficiency of photodegradation was found to increase rapidly with increasing the amount of the catalyst in the aqueous 25 ppm BPA samples. An increase in the catalyst concentration by 0.1% (w/w) accelerated the photodegradation efficiency of BPA by 35% for SnO$_2$, and 26% for TiO$_2$, whereas only 6.7% were achieved for ZnO (Fig. 8). This can be explained by the higher UV absorption of ZnO and its capability to break the BPA compound more efficiently than TiO$_2$ and SnO$_2$ at even low concentration of the catalyst (Lizama et al., 2002; Poulios and Tsachpinis, 1999; Sakthivel et al., 2003). Furthermore, ZnO shows a higher capacity to degrade BPA by releasing free oxidative radicals (e.g. hydroxyl radicals •OH) into the aqueous suspension. Moreover, the results indicated that 95% of degradation could be achieved using ZnO after 3 hour of degradation, whereas only 63% and 57% for the other catalysts TiO$_2$ and SnO$_2$, respectively, under the same experimental conditions. Based on what was mentioned above, the subsequent decontamination approach was implemented using the lowest possible amount of the catalysts considering the preservation of physical and chemical condition of water, hence only 0.1% (w/w) of the catalysts was used.

### 3.5 Advanced photocatalytic oxidation

The experiments show a high dependency of photodegradation on the initial oxidant concentration. The efficiency of BPA photodegradation increases proportionally with the NaOCl concentration in aqueous samples. The results are presented in Table 1. The results also revealed that the oxidization resulted in a high degradation efficiency, and that a slight increase in the oxidant concentration had a marked stimulatory effect on the degradation of BPA. More than 90% of degradation was achieved





within less than 1 h using ZnO as catalyst at 0.3 mM NaOCl (Fig. 9). However, the concentration of NaOCl influenced the degradation process and significantly accelerated the disintegration reactions of BPA. For instance, with an increasing of the NaOCl concentration by 0.2 mM a higher degradation can be achieved (about 15- 20%).

In this study, two reaction phases were recognized using the PCO approach. The first was characterized by a sharp decline in

BPA concentration (in the first 15 min of the experiment), while in the second phase the efficiency of degradation stabilized over time and the reaction reached a steady state (no significant change in concentration).  This can be explained either by less generation of $e^-/h^+$ (valence electron/ hole in the valence band) which reduced the photodegradation efficiency (Konstantinou and Albanis, 2004), or less formed •OH and $O_2^{2-}$on the surface of the catalyst resulting in low relative ratios of •OH and $O_2^{-2}$ present for attacking BPA and thus accelerating the photodegradation (Lathasree et al., 2004). The process will continue until

complete oxidation is achieved (second part of reaction). Figure 10 shows that the ZnO suspension exhibits high performance to remove BPA from water samples. This is explained by its ability to absorb larger fractions of photon energy than the other photocatalysts. This property that is one of the most important advantages of using ZnO for this type of treatment (Pardeshi and Patil, 2008). The other degradation byproducts were investigated using GC-MS. The analyses indicated the presence of phenol-derivative products, butylated hydroxytoluene and chlorohydroquinone based on the library database of the National

Institute of Standard and Technology (NIST). Table 2 lists fragments (*m/z*) and its relative abundances (*%*) for the main two disintegration products. The disintegration process of BPA into different relevant phenolic products has been divided into 4 stages over time as shown in the Fig. 11.  The results show, that BPA reacts rapidly with sodium hypochlorite. It is likely that a chlorination dominates the degradation process by the electrophilic attack of HOCl on the phenoxide ions. Therefore, various chlorinated derivatives can be observed after 10min of degradation. This was observed by comparing the detected products in

GC-MS with the NIST database for each stage. However, the kinetic of degradation for endocrine compounds such as BPA is rather complex and requires further investigation.

### 3.5 Assessment of degradation approaches

The performance of decontamination approaches was evaluated through two key factors; treatment time at lowest amount of catalysts as well as NaOCl, and the achieved purification success. Figure 12, compares different methods used to remove BPA

from water at fixed illumination time of 1h and shows that photodegradation using ZnO and NaOCl as oxidizing agent is the best option to remove BPA from water. As previously noted, ZnO exhibited a higher capacity to disintegrate organic contaminants from water in the presence of UV irradiation and it offers the additional advantage of lower turbidity which was not the case for the other catalysts (Fig. 13).

### 4 Conclusion

Photodegradation of BPA using PCO with ZnO as catalyst is highly sensitive to UV intensity and the initial concentration of oxidant. ZnO showed distinctive properties and was very stable during the degradation process. The removal of BPA from





water was significantly enhanced by low doses of oxidant (NaOCl). The proposed remediation approach can be adopted for removal of BPA and phenolic products in natural water. More investigations are required to study the disintegration of BPA by photodegradation under various natural conditions at different temperature and pH and in the presence of different cations and anions. Also the effect of the treatment on the mineralization of natural aqueous system should be studied in order to

achieve the best ecological way to remove BPA and phenolic contaminants from natural water resources. A last point for further investigations is the utilization of the catalysts in a real world scenario and whether NaOCl can be replaced e.g. by ozone which could be produced by means of a solar driven ozone generator.

## 4 Acknowledgments

This work was supported by the German Academic Exchange Service (DAAD). Special thanks due the laboratory staff at the

TU Freiberg for supporting this research.

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

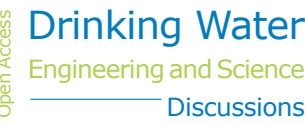

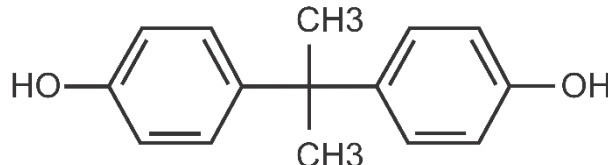

**Figure 1: Structure of Bisphenol A.**





**Figure 2: Photo-Fries rearrangement of BPA modified after A. Rivaton (1995).**





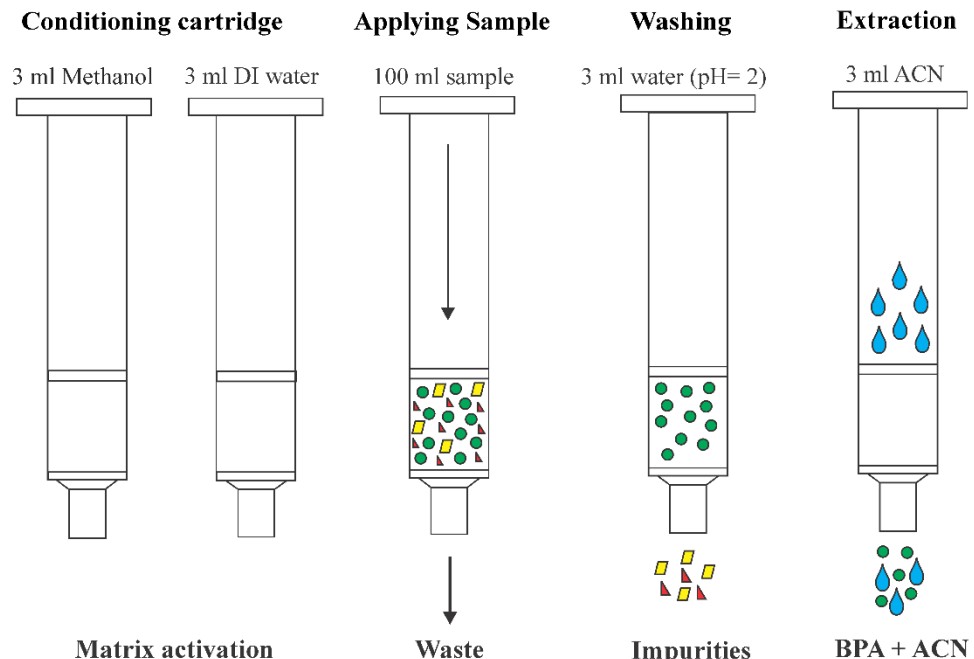

**Figure 3: The steps of solid phase extraction.**

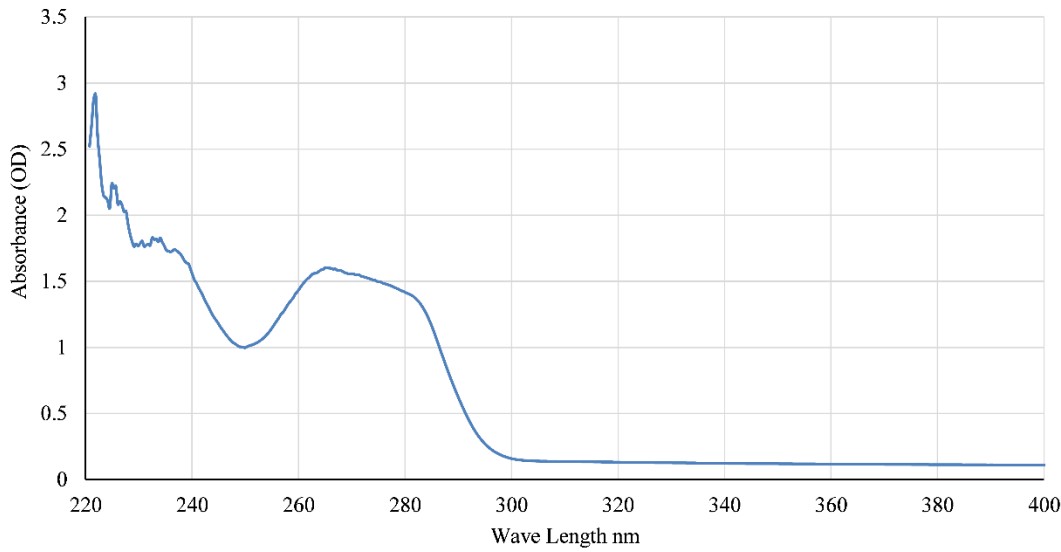

**Figure 4: Bisphenol A UV-VIS absorbance spectrum in water (OD: optical density).**





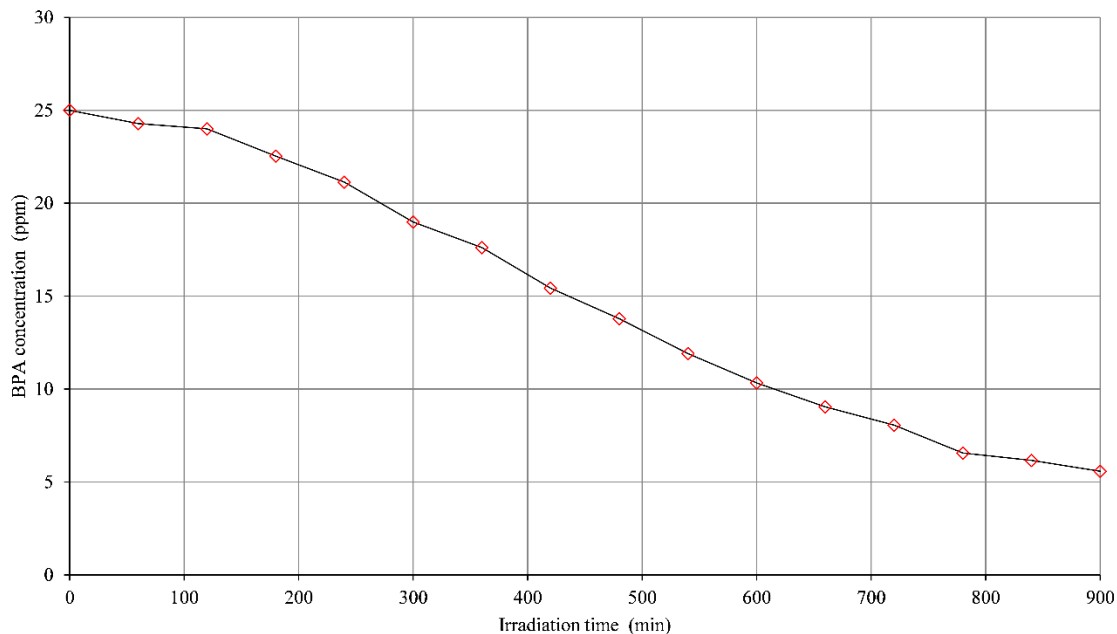

**Figure 5: Photodegradation of BPA in pure water samples under low intensity UV irradiation.**

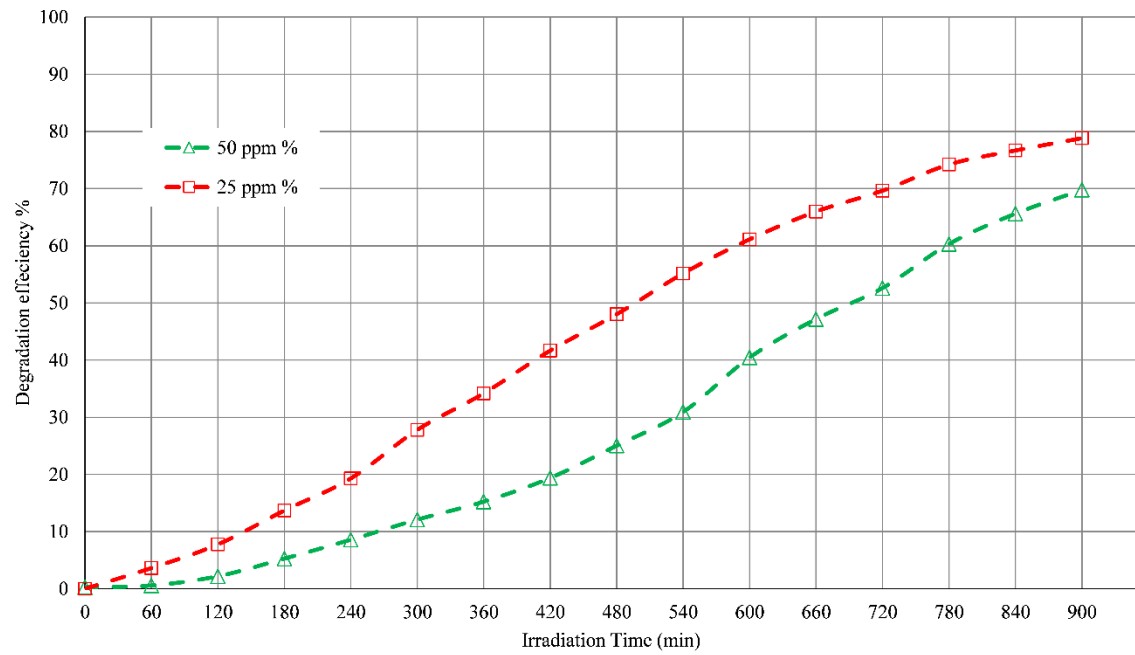

**Figure 6: The effect of the BPA initial concentration on the progress of photodegradation.**





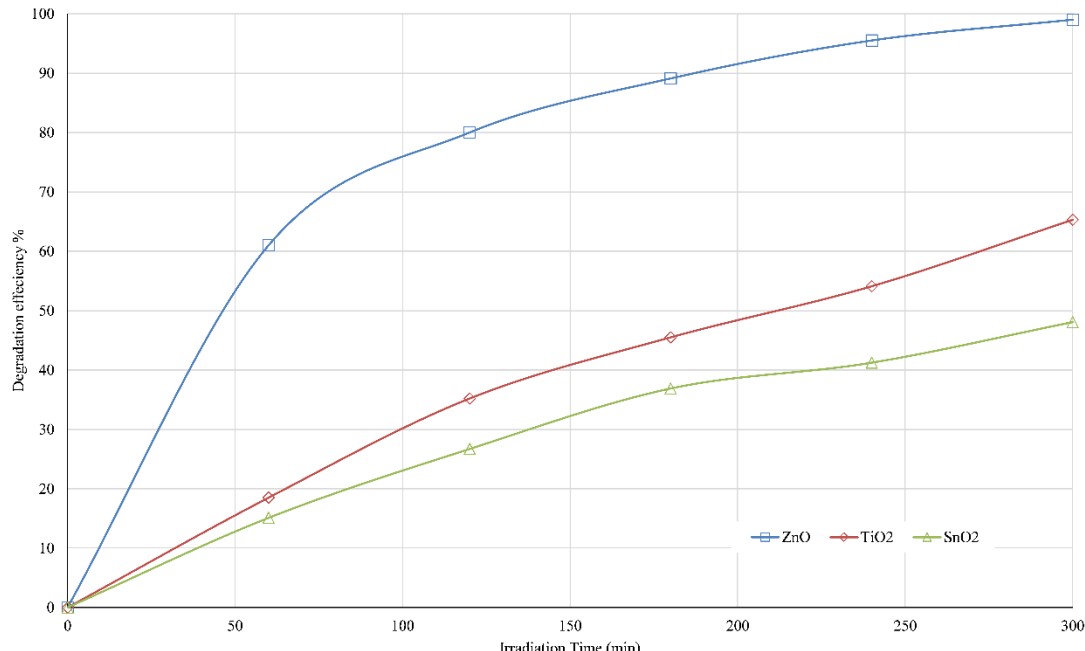

**Figure 7: Degradation efficiency of BPA in aqueous suspensions of ZnO, TiO₂ and SnO₂ under UV irradiation.**

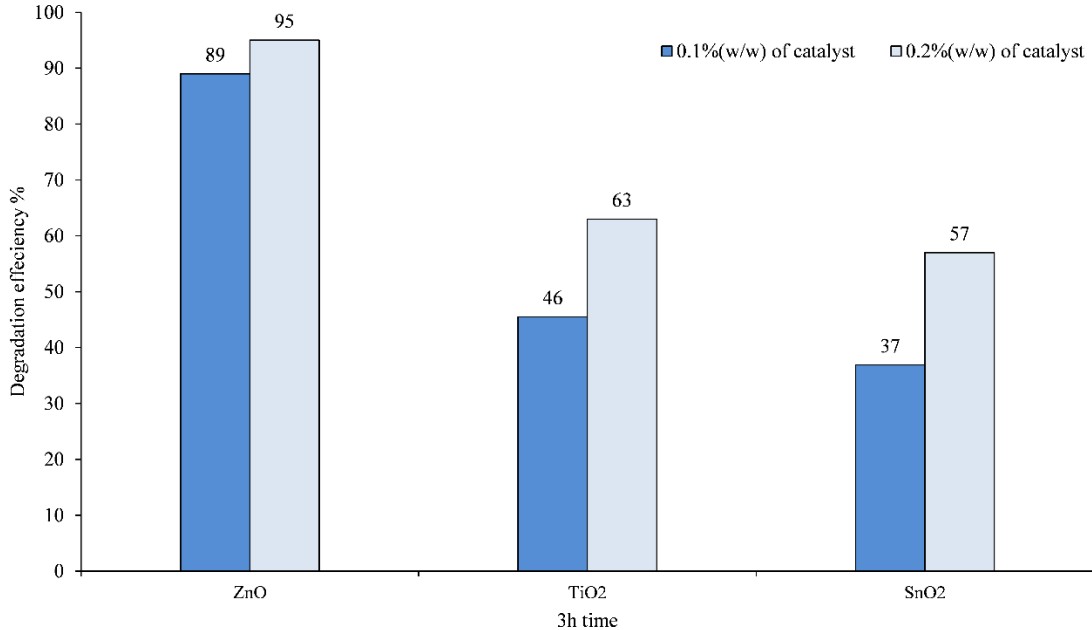

**Figure 8: The impact of catalyst concentration on the efficiency of photodegradation.**





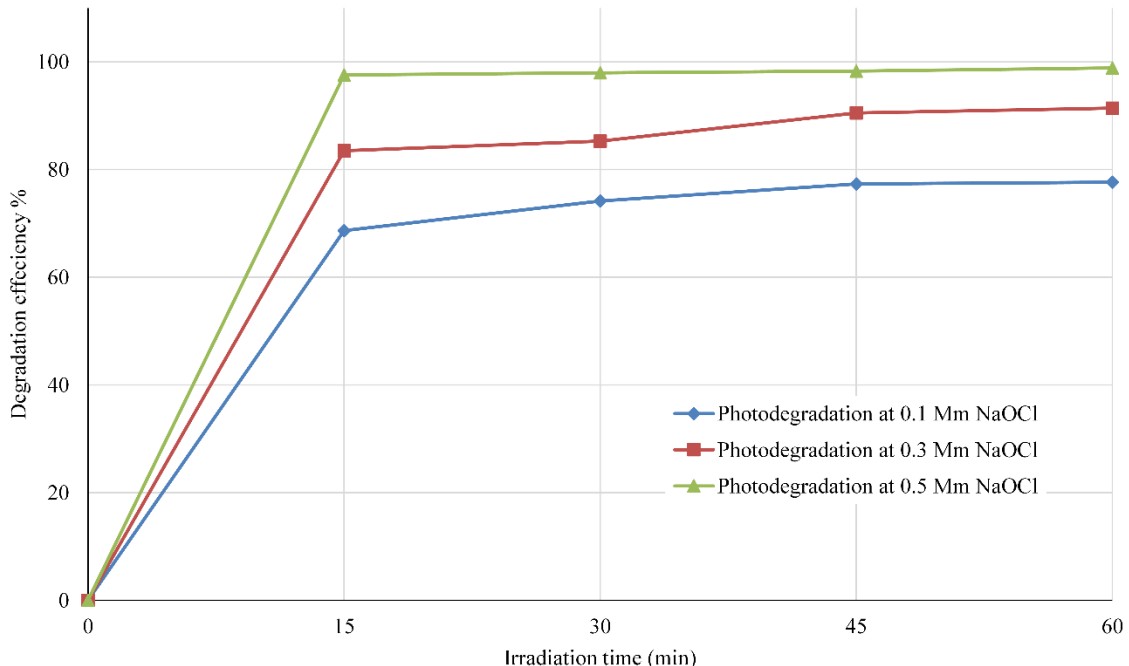

**Figure 9: The efficiency of the BPA degradation by different doses of NaOCl in suspensions of ZnO.**

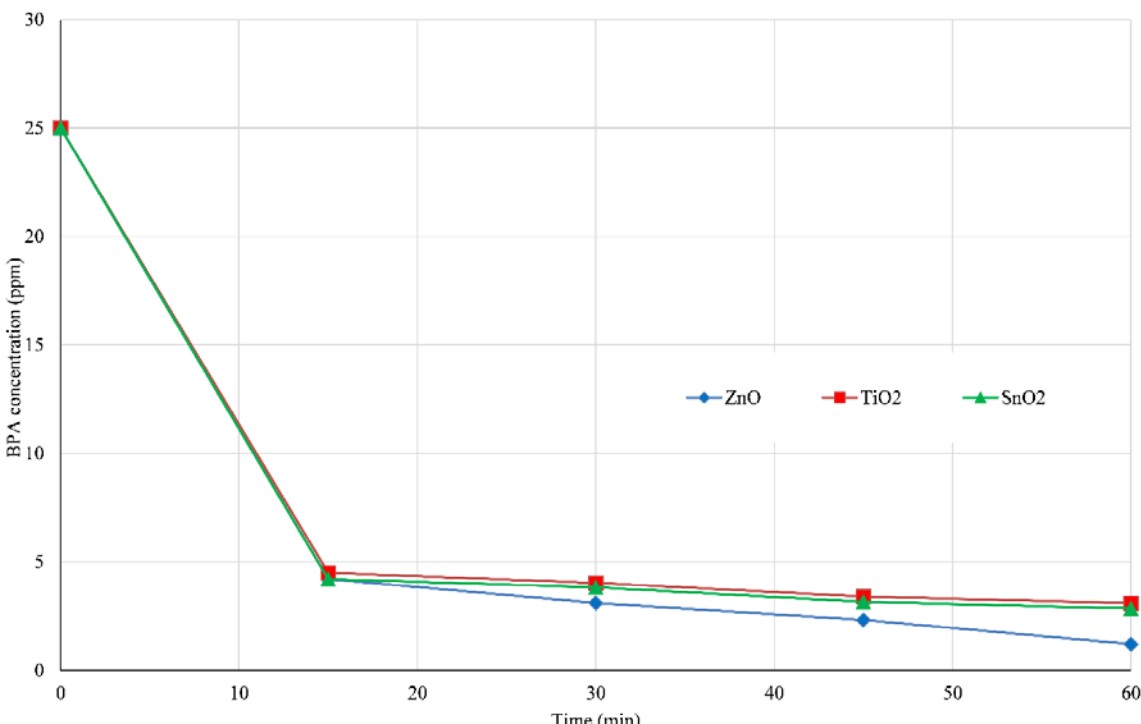

**Figure 10: Degradation efficiency of using ZnO, TiO₂, SnO₂ suspensions and advanced photooxidation (0.3 mM NaOCl).**




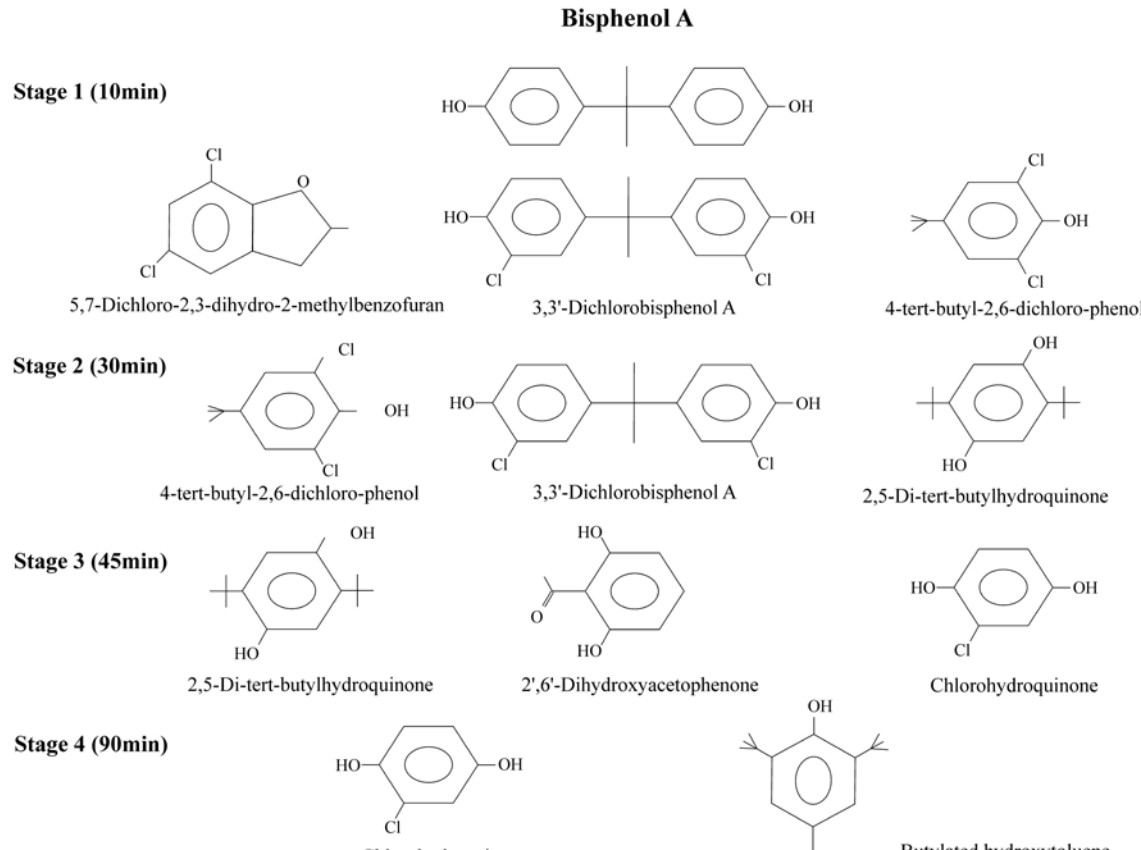

**Figure 11: Advanced photocatalytic oxidization of BPA and predicted degradation compounds in GC-MS of the BPA aqueous samples.**





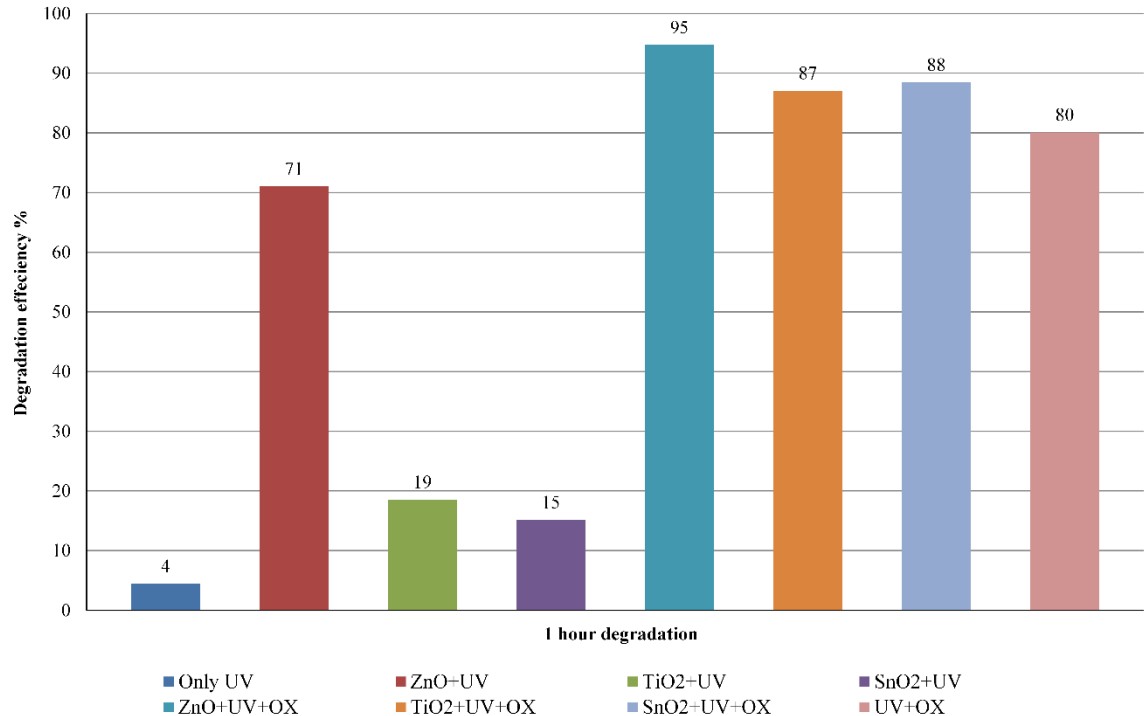

**Figure 12: Degradation performance of different used methods for removing BPA from water.**

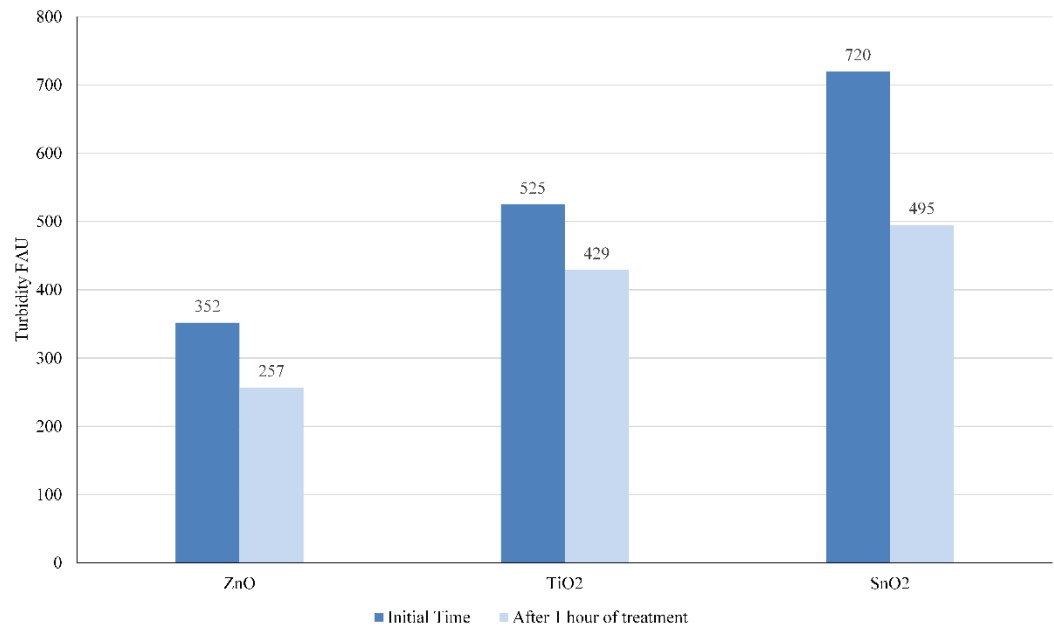

**Figure 13: The change in turbidity of aqueous suspensions of ZnO, TiO₂ and SnO₂ in the beginning and after 1h of treatment using the advanced oxidation method with NaOCl. (The turbidity index is given in formazin attenuation units FAU).**



**Table 1. The effect of different oxidant concentration on the photodegradation of BPA within 1 hour of irradiation.**

| Sample vol. (ml)/ conc. (ppm) | Catalyst 0.1 % ( w/w ) | Sodium hypochlorite mM | Degradation % |
|---|---|---|---|
| 100/ 25 | ZnO | 0.1 | 77.6 |
| | | 0.3 | 94.7 |
| | | 0.5 | 99.9 |
| 100/ 25 | $TiO_2$ | 0.1 | 56.1 |
| | | 0.3 | 87.7 |
| | | 0.5 | 98.1 |
| 100/ 25 | $SnO_2$ | 0.1 | 56.7 |
| | | 0.3 | 88.5 |
| | | 0.5 | 94.3 |

5   **Table 2. The final degradation products indicated by GC-MS.**

| Peak no. | Retention time (min) | m/z, (abundance % ) | Molecular weight m/z | Formula |
|---|---|---|---|---|
| 1 | 3.49 | 144, (47) | 144.56 | $C_6H_3(OH)_2Cl$ |
| 2 | 4.28 | 220, (98) | 220.34 | $C_{15}H_{24}O$ |