# Peer review of "Optimized photodegradation of Bisphenol A in water using ZnO, TiO2 and SnO2 photocatalysts under UV radiation as a decontamination procedure"

_Drinking Water Engineering and Science, 2016_

## Referee Comment (RC1) · Rudy Abo et al. · 13 Jul 2016

A.H. Knol (Referee)

This is an interesting study relying on photo degradation of Bisphenol A with and without different catalysts. It does address an important issue in water treatment. Still, the manuscript could benefit from the following considerations and remarks:

Page 2, line 20: The statement "Metal oxides have been widely used as catalysts for Photo degradation in recent years" asks for recent references.

[Figure]

Page 3, line 3: If oxygen is a strong oxidant, why not first saturate the water with oxygen before applying photolysis?

Page 3, line 31: Explain why these lamps with these wavelengths and intensity range mimics solar radiation. This is an important issue regarding the objective of the study (P1, line 8). Is the unit $\mu W/m^3$ correct?

P4, line 2: During the experiments of 15 hours, every hour 5 ml is extracted from the sample. Did that effect the UV-dose of the remaining sample and could that effect the results?

Page 4, line 9: "BPA in 100 ml ultra-pure water were prepared and mixed for 15 min using a magnetic stirrer at 400 rpm rotation speed in order to allow maximal sorption of BPA on the catalysts' surface". The samples were not stirred during irradiation? Did the catalyst settle and did that effect the experiments?

Page 5, line 23: Can you explain why the photo-Fries reaction rearrangement of BPA starts after about 240 minutes?

P5, line 26: Do you have an explanation for the concentration dependency in degradation rate?

P6, line 7&8: Does the value of the gap has a unit?

P6, line 18: The mentioned percentages do not match with Fig. 8. If the numbers in Fig. 8 are correct, than the relative increase in degradation is the same for SnOâĆĆ and ZnO, namely about 55%. Could that influence your conclusions about this experiment?

Page 7, line 13: "The other degradation byproducts were investigated using GC-MS". Which byproduct(s) was(were) discussed earlier in this paragraph? For readability it is to consider to start a new paragraph about byproducts.

Page 7, line 1: The applied NaOCl doses were high, till 37.5 mg/L. Do you agree that, depending on the water quality of surface and groundwater, a considerable amount of

DBPs can be formed, also with toxic properties (as BPA). To avoid that the "remedy is worse than the disease", could a footnote be of importance?

Page 7, line 17: "The results show, that BPA reacts rapidly with sodium hypochlorite. It is likely that a chlorination dominates the degradation process by the electrophilic attack of HOCl on the phenoxide ions". This stat ement seems in conflict with the statement in the Introduction (Page 2, line 33). Can you comment?

Page 7, line 18: Which part of the degradation efficiency of BPA is caused by the reaction with HOCl and which part by the catalyst?

Please also note the supplement to this comment:
http://www.drink-water-eng-sci-discuss.net/dwes-2016-5/dwes-2016-5-RC1-supplement.pdf

---

## Referee Comment (RC2) · Anonymous Referee #2 · 19 Jul 2016

Drinking Water Engineering and Science Discussions, manuscript #: doi:10.5194/dwes-2016-5, 2016 Manuscript Title: Optimized photodegradation of Bisphenol A in water using ZnO, TiO2 and SnO2 photocatalysts under UV radiation as a decontamination procedure Authors: Rudy Abo, Nicolai-Alexeji Kummer, Broder J. Merkel

This paper deals with optimizing the treatment procedure accelerating the photodegradation process of BPA and other phenolic compounds in natural water by using various photodegradation approaches; photodegradation, photo-oxidation/ photocatalytic

degradation and advanced photocatalytic oxidation degradation. To develop an improved technique that can be used as remediation procedure for BPA contaminated surface and groundwater based on solar radiation, experiments were performed under low-intensity UV mimics natural solar irradiation. The topic of the manuscript is interesting. However, there are some concerns and some aspects that need more clarity. In addition, more detail explanation of some of the results presented is needed in the paper. Therefore, the manuscript has to be revised due to concerns cited below:

Major comments: Technical comments 1)The intensity VS. the light wavelength of the light source that mimics natural solar light should be illustrated.

2)Page 2, Line 32. The author explained that "the advanced photocatalytic oxidation using sodium hypochlorite (NaOCl) as an oxidizing agent can accelerate the degradation efficiency by releasing oxygen (O2) into the water". The phenomenon is interesting. However, the mechanism is not clear. More experiment should be designed to reveal the mechanism. Is any role of chlorine play in the reaction?

3)Page 3, Line 6. The author said that "Few studies have dealt with other catalysts such as ZnO and SnO2, particularly the ZnO which shows high stability and a large band gap in comparison to other existing catalysts." Please provide relevant supporting papers? To the best of my knowledge, the stability of ZnO is not as good as TiO2.

4)Page 5, Section 3.2 and 3.3. Just suggest. Section 3.2 and 3.3 were conducted in the same conditions and I think Fig.4 just looks like one part of Fig.5. They can be showed just in one Figure. (Degradation efficiency of BPA and downtrend of BPA concentration (ppm) just express the same meaning.)

5)Page 7, Line27. The author said that "it offers the additional advantage of lower turbidity which was not the case for the other catalysts (Fig. 13)." What exactly did it express? For good water quality or easy separation of ZnO? Please clarify it more clearly.

Please also note the supplement to this comment:

http://www.drink-water-eng-sci-discuss.net/dwes-2016-5/dwes-2016-5-RC2-supplement.pdf

---

## Referee Comment (RC3) · Anonymous Referee #3 · 22 Jul 2016

Page 5 Line 28. It was mentioned that the lower initial concentration of BPA employed higher degradation rate. Once the Y axis value were changed from degradation efficiency (%) to BPA concentration (ppm), the results will be different. How to explain it?

Page 6 Line 6. It is obvious from Figure 7 that ZnO is a more effective catalyst than $TiO_2$ and $SnO_2$. The band gap was posed as an explanation. How about other mechanisms? It is better to find more theories from literature as support.

Page 7 Line 10. It is mentioned that ZnO is a better catalyst in photo-oxidation process and it adsorbs more photon-energy than the other photocatalysts. However, as displayed in Figure 10, the advantage of ZnO over TiO2 and SnO2 were insignificant. Since no duplicate experiment was carried out, the conclusion might be marginal. If the results were accurate, more explanations about mechanisms need to be discussed further.

It will be better if the results without catalysts can be shown in Figure 7, 9 and 10.

Page 7 Line 18. Electrophilic attack of HOCl on the phenoxide ions was raised as the mechanism of oxidation. In combination with Figure 11, more details can be clarified, e.g. how does the HOCl attack BPA and the degradation compounds step by step; which positions/sites are preferred by HOCl-attacking.

Page 7 Line 31. There was no result which showed ZnO is very stable during the degradation process. Please cite reference articles.

---

## Author Comment (AC1) · 24 Jul 2016

Dear Mr. A. H. Knol, We are very grateful to you for your constructive comments and suggestions, which will certainly improve the quality of our manuscript. Specific questions are responded one by one below:

1- Page 2, line 20: The statement "Metal oxides have been widely used as catalysts for Photo degradation in recent years" asks for recent references. Authors: new reference added to the manuscript.

2- Page 3, line 3: If oxygen is a strong oxidant, why not first saturate the water with

oxygen before applying photolysis?

Authors: The author agree. We can expect significant acceleration of the photodegradation process by pre-saturation of contaminated water with oxygen. The combination of different techniques such as ozonation could be also more efficient than ordinary approach. Thus, we suggested to use of ozone as disinfection/ oxidation agent instead of NaOCI in the future.

3- Page 3, line 31: Explain why these lamps with these wavelengths and intensity range mimics solar radiation. This is an important issue regarding the objective of the study (P1, line 8). Is the unit  $\mu$ W/m3 correct?

Authors: The solar ultraviolet UV-C, UV-B and UV-A wide range exposure are difficult to achieve in the lab (100-280, 280-315, and 315-400 nm, respectively). Considering the fact, that UV wavelength below 200 nm exists only in the vacuum, and the effective UV spectrum (250-340 nm), so we assumed UV irradiation source between 254 and 365 nm can simulate more or less the solar UV irradiation. This statement was added to the text. Regarding the unit of UV- intensity, the correct unit is  $\mu$ W/cm2

4- P4, line 2: During the experiments of 15 hours, every hour 5 ml is extracted from the sample. Did that effect the UV-dose of the remaining sample and could that effect the results? Authors: Interesting question! Unfortunately, the effect of residual amount of the contaminated water under the same UV- exposure wasn't investigated.

5- Page 4, line 9: "BPA in 100 ml ultra-pure water were prepared and mixed for 15 min using a magnetic stirrer at 400 rpm rotation speed in order to allow maximal sorption of BPA on the catalysts' surface". The samples were not stirred during irradiation? Did the catalyst settle and did that effect the experiments?

Authors: The suspension was stirred at 250 rpm during the photocatalytic process to prevent catalyst- deposition and ensure the continuity of degradation. With concern to that, new paragraph was added to the text explaining the mentioned point.

6- Page 5, line 23: Can you explain why the photo-Fries reaction rearrangement of BPA starts after about 240 minutes?

Authors: The adsorption of shortwave UV irradiation of 254 nm induces disintegration of CO-O bond forming two main derivative compounds as pheynlsalicylate and dihydroxybenzophenone. The high concentration of dihydroxybenzophenone near the surface may slow the degradation as efficient UV absorber and delayed the breakthrough of BPA. However, the long photolyses-time requires more investigation by means of FTIR spectroscopy.

7- P5, line 26: Do you have an explanation for the concentration dependency in degradation rate?

Authors: A great number of studies have linked the initial concentration and the photodegradation rate of polycarbonate compounds, which can be described in terms of time as follows: Degradation rate =  $-\partial C/\partial t$ ; where  $\partial C$  is the relative change in concentration The lower the initial concentration the higher the degradation rate since photo-oxidation of POC will work faster at lower amount of contaminants. The degradation dependency on the initial concentration is described by the kinetic model of Langmuir-Hinshelwood (Chen and Ray, 1998; Poulios and Tsachpinis, 1999). In this study the photodegradation rate % was calculated using the following equation: Photodegradation: P(C-C)/C0

8- P6, line 7&8: Does the value of the gap has a unit?

Authors: The band gap unit is eV (Electron volt) updated in the corresponding paragraph.

9- P6, line 18: The mentioned percentages do not match with Fig. 8. If the numbers in Fig. 8 are correct, than the relative increase in degradation is the same for SnOâĆĆ and ZnO, namely about 55%. Could that influence your conclusions about this experiment?

Authors: We agree the reviewer. The percent increase was miscalculated for the SnO2 and TiO2. The percent increase was simply calculated as follows: Percent increase in degradation efficiency= [(new concentration – original concentration)/ original concentration]\*100. Thus the relative percent increase the degradation efficiency is 54, 37 and 6.7% for the catalyst SnO2, TiO2 and ZnO, respectively. The changes were compared to the previous version of the text (Results, and conclusion) and there are no conflict by the interpretation. We proofed the corresponding paragraph for better reading.

10- Page 7, line 13: "The other degradation byproducts were investigated using GC-MS". Which byproduct(s) was(were) discussed earlier in this paragraph? For readability it is to consider to start a new paragraph about byproducts.

Authors: Degradation byproducts was discussed briefly in the induction in terms of Photo-fries reactions. In any case, tracing of resulting derivatives and the kinetic of disintegration during the advanced photocatalytic oxidation requires further investigation. We agree the reviewer. A new paragraph added to the manuscript entitled: Byproducts of photocatalytic oxidation.

11- Page 7, line 1: The applied NaOCI doses were high, till 37.5 mg/L. Do you agree that, depending on the water quality of surface and groundwater, a considerable amount of DBPs can be formed, also with toxic properties (as BPA). To avoid that the "remedy is worse than the disease", could a footnote be of importance?

Authors: The authors agree the reviewer. Yes! 0.5 or even 0.3 mM is reality high dosage of NaOCI, particularly if we considered that sodium hypochlorite is a dangerous and corrosive oxidizing material. Thus we recommended to use it at minimal effective concentration of 0.1 mM or less (e.g. 0.05 nM). Advantage and disadvantage of using NaOCI were added to the modified version of the paper as well as the recommended dosage.

12- Page 7, line 17: "The results show, that BPA reacts rapidly with sodium hypochlorite. It is likely that a chlorination dominates the degradation process by the electrophilic attack of HOCI on the phenoxide ions". This stat ement seems in conflict with the statement in the Introduction (Page 2, line 33). Can you comment?

Authors: Yes! At elevated concentration of NaOCI

13- Page 7, line 18: Which part of the degradation efficiency of BPA is caused by the reaction with HOCI and which part by the catalyst?

Authors: The oxidation process with HOCI dominate the first part of the reaction and advanced by the catalyst effect over the second stage. Change made to the corresponding paragraph to clarify that point.

---

## Author Comment (AC2) · 31 Jul 2016

Dear Referee 2, The authors would like to thank you for your constructive comments on our manuscript. We will consider your remarks to improve this work. Please find below our response to your comments:

1) The intensity VS. the light wavelength of the light source that mimics natural solar light should be illustrated.

We would like to clarify, that the exposed UV the used lamps replicates to some extent the electromagnetic spectrum UVA and UVC with a wavelength between 254 and 365

nm, and not the entire spectrum of the sun light from 10 to 400 nm. However, the UV radiation between 254-360 nm has the crucial role in the disintegration of most organic compounds. We clarified this point in the abstract and throughout the text. Unfortunately, laboratory and industrial photometric data about the lamps aren't available.

2) Page 2, Line 32. The author explained that "the advanced photocatalytic oxidation using sodium hypochlorite (NaOCl) as an oxidizing agent can accelerate the degradation efficiency by releasing oxygen (O2) into the water". The phenomenon is interesting. However, the mechanism is not clear. More experiment should be designed to reveal the mechanism. Is any role of chlorine play in the reaction?

The authors thanks the reviewer for his valuable remark. The mentioned paragraph was included contain uncertain information due to narrative order. The paragraph was revised in the new version of the manuscript, including the written equation and cited researches. The statement about releasing of oxygen occurs during the photocatalytic process was removed. The role of chlorine explained in more details.

3) Page 3, Line 6. The author said that "Few studies have dealt with other catalysts such as ZnO and SnO2, particularly the ZnO which shows high stability and a large band gap in comparison to other existing catalysts." Please provide relevant supporting papers? To the best of my knowledge, the stability of ZnO is not as good as TiO2.

Despite the widespread use of TiO2 in the photodegradation of organic compounds, many published studies beside this work showed higher degradation efficiency by using ZnO2. However, other studies argued that sometimes ZnO appears to be unstable due to the dissolution in water. These statements requires more investigation and further research. We added were added the corresponding references to the manuscript.

4) Page 5, Section 3.2 and 3.3. Just suggest. Section 3.2 and 3.3 were conducted in the same conditions and I think Fig.4 just looks like one part of Fig.5. They can be showed just in one Figure. (Degradation efficiency of BPA and downtrend of BPA concentration (ppm) just express the same meaning.)

The referee meant Fig. 6 and 7 since Fig. 4 states totally different information (UV-VIS absorbance curve). However, providing 2 figure will reinforce the idea of each section (change in concentration by photolysis and the effect of BPA initial concentration), thus we suggest to leave them as they are.

5) Page 7, Line27. The author said that "it offers the additional advantage of lower turbidity which was not the case for the other catalysts (Fig. 13)." What exactly did it express? For good water quality or easy separation of ZnO? Please clarify it more clearly.

This was intended to easily removal of ZnO particles from treated water after completion of photocatalytic oxidation process. We reworded the mentioned paragraph above to clarify the idea.

---

## Author Comment (AC3) · 31 Jul 2016

Dear Referee 3, The authors would like to thank you for your constructive comments on our manuscript. Please find below our response to your comments: 1) Page 5 Line 28. It was mentioned that the lower initial concentration of BPA employed higher degradation rate. Once the Y axis value were changed from degradation efficiency (%) to BPA concentration (ppm), the results will be different. How to explain it?

The relative percent increase of the degradation (%) over time is a function of the change in the concentration and can be written as follows:

[Figure]

Degradation efficiency % = [C0-Ct/C0]*100 ; where C0 is the initial concentration and Ct is the concentration of BPA at specific time. Thus, the degradation efficiency will increase over time and wise versa for the concentration. Please have a look at the provided figure (Fig.1).

2) Page 6 Line 6. It is obvious from Figure 7 that ZnO is a more effective catalyst than TiO2 and SnO2. The band gap was posed as an explanation. How about other mechanisms? It is better to find more theories from literature as support.

The authors improved the corresponding paragraph by providing more information about the mechanism of degradation, ZnO photocatalyst properties with supporting literatures.

3) Page 7 Line 10. It is mentioned that ZnO is a better catalyst in photo-oxidation process and it adsorbs more photon-energy than the other photocatalysts. However, as displayed in Figure 10, the advantage of ZnO over TiO2 and SnO2 were insignificant. Since no duplicate experiment was carried out, the conclusion might be marginal. If the results were accurate, more explanations about mechanisms need to be discussed further.

The authors agree with referee. The mechanism of degradation was considered in more details throughout the new version of the manuscript.

4) It will be better if the results without catalysts can be shown in Figure 7, 9 and 10.

Results of degradation in absence of catalysts were added to the figure 7. Unfortunately, the results without catalysts can't be plotted on the figures 9 and 10 due to the differences in time resolution and missing measurements without catalysts at each 15 min time-interval for this stage of APO. However, Fig. 12 compares different used approaches.

5) Page 7 Line 18. Electrophilic attack of HOCl on the phenoxide ions was raised as the mechanism of oxidation. In combination with Figure 11, more details can be clarified,

e.g. how does the HOCl attack BPA and the degradation compounds step by step; which positions/sites are preferred by HOCl-attacking.

Authors agree the referee, more details added to the corresponding paragraph.

6) Page 7 Line 31. There was no result which showed ZnO is very stable during the degradation process. Please cite reference articles.

The mentioned paragraph was modified with supporting references.

———————————————————

[Figure]

[Figure]

Fig. 1.